# Antibiofilm Activity on *Candida albicans* and Mechanism of Action on Biomembrane Models of the Antimicrobial Peptide Ctn[15–34]

**DOI:** 10.3390/ijms21218339

**Published:** 2020-11-06

**Authors:** Francisca Lidiane Linhares de Aguiar, Nuno C. Santos, Carolina Sidrim de Paula Cavalcante, David Andreu, Gandhi Radis Baptista, Sónia Gonçalves

**Affiliations:** 1Laboratory of Biochemistry and Biotechnology, Institute for Marine Sciences, Federal University of Ceará, Fortaleza, Ceará 60165-081, Brazil; lidianelinhares@yahoo.com.br; 2Graduate Program in Pharmaceutical Sciences, School of Pharmacy, Dentistry and Nursing, Federal University of Ceará, Fortaleza, Ceará 60430-170, Brazil; 3Instituto de Medicina Molecular, Faculdade de Medicina, Universidade de Lisboa, 1649-028 Lisbon, Portugal; nsantos@fm.ul.pt; 4Center for Science and Technology, State University of Ceará, Fortaleza, Ceará 60714-903, Brazil; carolsidrim81@gmail.com; 5Department of Experimental and Health Sciences, Universitat Pompeu Fabra, Barcelona Biomedical Research Park, 08003 Barcelona, Spain

**Keywords:** crotalicidin, venom-derived peptide, antimicrobial peptide, membrane-active peptide, *Candida albicans*, biofilm, yeast protoplast, biomembrane

## Abstract

Ctn[15–34], the C-terminal fragment of crotalicidin, an antimicrobial peptide from the South American rattlesnake *Crotalus durissus terrificus* venom, displays remarkable anti-infective and anti-proliferative activities. Herein, its activity on *Candida albicans* biofilms and its interaction with the cytoplasmic membrane of the fungal cell and with a biomembrane model in vitro was investigated. A standard *C. albicans* strain and a fluconazole-resistant clinical isolate were exposed to the peptide at its minimum inhibitory concentration (MIC) (10 µM) and up to 100 × MIC to inhibit biofilm formation and its eradication. A viability test using XTT and fluorescent dyes, confocal laser scanning microscopy, and atomic force microscopy (AFM) were used to observe the antibiofilm effect. To evaluate the importance of membrane composition on Ctn[15–34] activity, *C. albicans* protoplasts were also tested. Fluorescence assays using di-8-ANEPPS, dynamic light scattering, and zeta potential measurements using liposomes, protoplasts, and *C. albicans* cells indicated a direct mechanism of action that was dependent on membrane interaction and disruption. Overall, Ctn[15–34] showed to be an effective antifungal peptide, displaying antibiofilm activity and, importantly, interacting with and disrupting fungal plasma membrane.

## 1. Introduction

After the first report of the presence of cecropins in the hemolymph of *Hyalophora cecropia* moths [1], in 1981, reports followed of mammalian α-defensins in human neutrophils [2] and magainins in the skin secretions of the amphibian *Xenopus laevis* four years later [3]. The expression of antimicrobial peptides (AMPs) in organisms of all kingdoms has been recognized as an essential feature of the innate immunity system, and hundreds of AMP sequences belonging to different structural and functional families have been characterized. Dedicated databases maintain information regarding this class of biologically active peptides in either native, synthetic, or de novo-designed versions [4,5,6]. More recently, AMPs from animal venoms have gained attention [7,8,9,10].

The vipericidins are a group of highly conserved cathelicidin-related antimicrobial peptides (CRAMPs) from the venom glands of South American pit vipers, exemplified by lachesicidin, lutzicidin, batroxicidin, and crotalicidin, as disclosed and characterized by some of us [11]. Vipericidins share also a high homology with CRAMPs from Asian elapid snakes, despite the fact that both groups of snakes (pit vipers and elapids) are phylogenetically and geographically distant relatives [12]. One of the best studied vipericidins, crotalicidin (Ctn), a linear 34-residue peptide (KRFKKFFKKVKKSVKKRLKKIFKKPMVIGVTIPF), rich in basic (Lys, Arg) residues and with an α-helical fold, is generated from a precursor made up of a signal peptide and a cathelin domain [1,4]. Its short N- and C-terminal fragments, Ctn[1–14] (KRFKKFFKKVKKSV) and Ctn[15–34] (KKRLKKIFKKPMVIGVTIPF), respectively, result from a careful structural dissection of the full-length Ctn and have been studied for anti-infective (antimicrobial, anti-parasite, and antiviral) and anti-proliferative properties (antitumoral), as reported elsewhere [13,14]. The two fragments adopt distinct folds in solution and display biological activities different from the parental Ctn; thus, Ctn[1–14] preserves helicity but, paradoxically, has no noticeable activities, while Ctn[15–34] adopts a mainly random structure and it is fully active as an AMP against Gram-negative bacteria, being less cytotoxic to healthy cells, highly stable in serum, as well as moderately active against several cancer cell lines in vitro [15,16]. Ctn[15–34] exerts its microbicidal effect by membrane disruption and interaction with microbial DNA, with a rapid onset of action, as observed for bacteria [17]. Recently, the antifungal properties of these peptides were demonstrated against planktonic forms of standard and clinical strains of pathogenic yeasts and dermatophytes, both susceptible and resistant to antifungals used in clinical settings, like polyenes (e.g., amphotericin B) and azoles (e.g., fluconazole) [18,19]. In this first study, it was demonstrated that Ctn[15–34] disrupts the plasma membrane integrity of *Candida* cells and induces apoptosis-mediated cell death [20]. These findings are interesting, given that systemic fungal infections cause some 1.5 million deaths each year worldwide [21]. The pathogenic fungi mainly responsible for systemic infections belong to the genus *Candida* and cause candidiasis with higher incidence in immunosuppressed patients [22]. Notably, these opportunistic pathogens are part of the human microbiota, colonizing the oral cavity, the gastrointestinal tract, and the vaginal mucosa [23]. The severity of candidiasis is associated to immune conditions of human host and virulence factors of this eukaryotic pathogen, such as dimorphism, expression of adhesion proteins, production of hydrolytic enzymes, and alteration of cell wall, which invariably exacerbate *Candida* pathogenicity [24]. Particularly, adhesins on the wall of *Candida* cells contribute to the formation of biofilms on general surfaces and on medical devices that are in contact with or implanted in patients, and such capacity to form biofilms was associated with invasiveness [25]. Biofilms are complex networks of a single or more species of microorganisms, surrounded by an extracellular polymeric matrix, consisting of carbohydrates, proteins, and nucleic acids. This structured consortium of cells provides some advantages for microorganisms, such as protection against environmental stress, deluding the host immune system, and, importantly, increasing resistance to antimicrobial drugs [25,26]. In fact, resistance to antifungals in biofilms is mainly attributed to the extracellular matrix acting as a shield that delays or prevents drug diffusion to individual cells located deep within a microcolony, therefore reducing the adsorption and/or neutralizing the drug effect [26,27].

Since antimicrobial peptides are poised as potential alternatives to conventional anti-infective chemotherapeutics in fighting infections and may be developed into biopharmaceuticals, in the present study, we evaluated the antifungal effects of Ctn[15–34] on planktonic (free-floating) cells and biofilms of two strains of *Candida albicans*: a reference (wild-type) drug-sensitive strain and a fluconazole-resistant clinical isolate. To this aim, biophysical methodologies were combined to address the fungicidal efficacy of Ctn[15–34] in *C. albicans* biofilms, as well as the candidacidal mechanism of action with lipid membranes that simulate the *Candida* cytoplasmic membrane. By combining fluorescence spectroscopy, dynamic light scattering, zeta potential assays, confocal microscopy, and atomic force microscopy, the membrane selectivity is demonstrated and new insights on the mechanism of action of Ctn[15–34] against *C. albicans* are revealed.

## 2. Results

### 2.1. Ctn[15–34] Activity against C. albicans Biofilms

After the synthesis, purification, and essential biochemical characterization of Ctn[15–34], antifungal activity was confirmed. Ctn[15–34] minimum inhibitory concentrations (MICs) for both *C. albicans* strains were in agreement with those previously determined. *C. albicans* breakpoints available from the European Committee on Antimicrobial Susceptibility Testing (EUCAST) and Clinical & Laboratory Standards Institute (CLSI) guidelines were used as parameters to establish yeast resistance [28]. For Ctn[15–34], a MIC of 10 μM (23.7 µg mL^−1^) was found for both strains tested (Table 1). According to the CLSI and EUCAST breakpoints, *C. albicans* (LABMIC 0125) was resistant to fluconazole (FCZ), with a MIC of 8 μg mL^−1^ (25 µM), higher than reported for this antifungal (4 μg mL^−1^ or 12.48 µM) [29], while for the standard wild-type strain (ATCC 90028), the MIC for FCZ is almost 10-fold lower, i.e., 1 μg mL^−1^ (3.12 µM).

To determine the antibiofilm effect of Ctn[15–34], the best conditions for biofilm formation were established (Appendix A). For the drug-resistant *Candida* isolate, the best time span for biofilm formation corresponds to 72 h, with a concentration of 10^7^ CFU mL^−1^. The standard strain forms biofilms at the same time interval as the drug-resistant isolate, but with a lower number of cells (10^4^ CFU mL^−1^). After establishing the best time and cell number to achieve biofilm formation of both strains, the next step was to evaluate the inhibitory activity of Ctn[15–34] on biofilm formation and biofilm eradication, in this later case, once the *Candida* biofilms were established. For that, a peptide concentration corresponding to the MIC (10 µM), as previously determined for the planktonic cells, was used. The peptide effects on biofilms were visualized by confocal microscopy (Figure 1a–p). After treatment with Ctn[15–34], cell density was clearly lower for the wild-type *C. albicans* ATCC 90028, when compared to the drug-resistant strain. This behavior was observed both in the assays of biofilm inhibition and eradication (Figure 1a–t). In the inhibition assay, the concentrations of 10 μM (MIC) and 100 μM (10 × MIC) considerably reduced the biofilm of the wild-type strain, whereas for the resistant strain, no change in cell number covering the support surface was observed. At 100 × MIC, there was a complete inhibition of biofilm formation of the susceptible strain. In the eradication assays, at concentrations lower than 100 × MIC, distinct results were observed when comparing the sensitive and the resistant clinical isolate strains. With the wild-type strain, the peptide completely eradicated the biofilm, whereas with the resistant strain, the biofilm was only partially eradicated, with some viable cells remaining (Figure 1). Illustrative xyz slice images of preformed *C. albicans* biofilm untreated or treated with Ctn[15–34] are presented in Appendix A. As shown, *C. albicans* biofilm densities are variable between both strains (Appendix A, control). After treatment with different Ctn[15–34] concentrations, a difference between strains in the action of this peptide was observed (Appendix A). Overall, treatment with Ctn[15–34] at 100 × MIC showed to be the best concentration to reduce and eradicate *C. albicans* biofilms.

As shown in Figure 1q–t, on the quantification of peptide-induced yeast cell death, the number of dead cells was low (approximately 2%) in biofilms of both *Candida* strains not exposed to Ctn[15–34], as expected. In the biofilm inhibition assay, at a peptide concentration equivalent to 100 × the MIC, the inhibition of biofilm formation and eradication were effective only for the drug-susceptible wild-type *Candida* strain. In contrast, such high concentration was effective in inhibiting biofilm formation with the fluconazole-resistant clinical strain, but eradication was not complete. Despite this, the number of dead cells continued to increase within 24 h of exposure to Ctn[15–34].

To further evaluate the inhibition and the eradication effects induced by Ctn[15–34] on *C. albicans* biofilms, atomic force microscopy (AFM) imaging, on intermittent contact mode, was used. Figure 2 highlights the main morphological changes that occurred in drug-resistant and wild-type *Candida* biofilms after exposure to Ctn[15–34]. As it can be noticed, Ctn[15–34] induced morphological alterations in a peptide concentration-dependent manner, even considering that untreated cells (controls) have some intrinsic distinctions: in the wild-type strain a smooth surface and regular shape were observed (Figure 2A,I), while in the drug-resistant *C. albicans* a light surface roughness, together with larger size and elongated shape, were normally visible (Figure 2E,M). On AFM images from the biofilm inhibition assay, the concentration-dependent irregularities in the cell surface of wild-type and drug-resistant *Candida* cells caused at 100 × MIC were evident (Figure 2B–D and Figure 2F–H, respectively). Moreover, the presence of pseudohyphae, which occurs when several elongated cells (blastoconidia) are joined together, was not observed. Additionally, on AFM images of wild-type *Candida* cells in the biofilm exposed to 100 × MIC, the predominant irregularities on the surface were accompanied by extravasation of intracellular content (Figure 2D). Images from the wild-type *Candida* cells in the biofilm eradication assay show that an irregular surface becomes evident with the increasing concentration of Ctn[15–34] (Figure 2J–L). Moreover, images from the biofilm along the eradication testing (Figure 2N–P) show apparently small vesicles on the surface of the drug-resistant cells when exposed to Ctn[15–34].

### 2.2. Evaluation of Ctn[15–34] Interaction with Lipid Membranes

Changes in dipole potential upon peptide-membrane interaction were assessed by labeling lipid vesicles and *C. albicans* cells with 4-(2-[6-(dioctylamino)-2-naphthalenyl]ethenyl)-1-(3-sulfopropyl)pyridinium inner salt (di-8-ANEPPS; Figure 3). By increasing the concentration of Ctn[15–34], no changes in the dipole potential of the lipid vesicles were observed. In contrast, Ctn[15–34] show measurable affinity with *C*. *albicans* cells (ATCC 90028), yielding an apparent dissociation constant (*K_d_*) of 14.5 ± 5.6 µM, indicating a preference (high affinity) of Ctn[15–34] for these cells.

To evaluate whether membranes properties are altered upon the interaction with Ctn[15–34], two biophysical parameters were evaluated by dynamic light scattering, namely, lipid vesicle hydrodynamic diameter (D_H_; Figure 4) and zeta potential (Figure 5). Ctn[15–34] did not promote any significant change in the D_H_ of 1-palmitoyl-2-oleoyl-*sn*-glycero-3-phosphocholine (POPC), POPC:Chol, and POPC:Erg (Figure 4). In contrast, using lipid vesicles with a composition that mimics the fungal cell membrane, Ctn[15–34] induced their aggregation at a peptide concentration of 50 μM (5 × MIC; Figure 4). Zeta potential results showed important differences between vesicles and cells. Despite the fact that the initial charge of the lipid vesicles of POPC, POPC:Chol, and POPC:Erg were slightly different, the increasing peptide concentrations did not promote additional changes in the zeta potential of these vesicles (Figure 5). Furthermore, no alteration in the zeta potential of *C. albicans* cells exposed to Ctn[15–34] (−4.32 mV) was observed.

Comparing zeta potential values for *C. albicans* cells with those for POPC:Erg, similarities can be found. Regardless of the inability to neutralize the surface charge, importantly, Ctn[15–34] induced a change in the zeta potential of the fungal membrane-like lipid vesicles from −25.1 ± 2.4 to −10.28 ± 2.6 mV (Figure 5). Antifungal agents may act on different cell components, but lipid membranes are commonly the first target to overcome. Thus, in addition to liposomes, the effect of Ctn[15–34] interaction with membranes and the neutralization of the membrane charge in *C. albicans* protoplasts is a robust evidence of the plasma membrane as the main initial target for antifungal activity. These cell-driven structures had an initial zeta potential of −12.37 ± 0.22 mV, which, after exposure to Ctn[15–34] (5 × MIC), increased to −5.9 ± 0.87 mV (Figure 5).

## 3. Discussion

The development of new antimicrobial drugs remains a major challenge to overcome the spread of drug resistance of clinically-relevant microbes, as strains are increasingly becoming less sensitive to conventional antibiotics. Previous studies have demonstrated that the crotalicidin fragment Ctn[15–34] is an effective antimicrobial peptide, with distinct anti-infective and anti-proliferative activities [13,14]. Its physicochemical properties, including small size, positive net charge, high hydrophobicity, and intermediate hydrophobic moment, endow Ctn[15–34] with a well-balanced structure-activity profile that makes it biologically active against bacteria and certain tumor cells, as well as stable in biological fluids [15,16]. In previous studies, we reported the antifungal properties of Ctn[15–34] against standard and clinical strains of susceptible and drug-resistant *Candida* species and dermatophytes, like *Cryptococcus* [18,19,20]. Here, we examined the antibiofilm potential of Ctn[15–34] on two *C. albicans* strains, including a drug-resistant clinical isolate, that are able to form biofilms. Moreover, the ability of Ctn[15–34] to interact with fungal membranes and biomembrane model systems was investigated as the initial mechanism of the peptide antifungal action. Firstly, the antifungal activity was confirmed, quantified as their minimum inhibitory concentrations against planktonic (free-floating) *C. albicans* cells of both a fluconazole-resistant clinical isolate and a standard wild-type ATCC strain. Both strains were shown to be susceptible to the fungicide activity of Ctn[15–34], with a MIC of 10 µM for both of them (Table 1).

Fungal biofilms differ from planktonic cells by their higher virulence, better adherence to mammalian tissue, and greater resistance to antifungals, especially azoles [30]. Although all *Candida* species are potentially prone to form biofilms, a well-recognized fact in clinical settings, the formation of biofilms depends on several factors, involving both the microorganisms themselves and the host [25]. Surfaces can play a determinant role on the formation of *Candida* biofilms [25,30]. Here, the biofilms of wild-type and fluconazole-resistant *C. albicans* were grown on polystyrene surfaces and differences in the amount of cells in the initial inoculum necessary to establish the biofilm were evidenced (Appendix A). As observed, despite the cell number- and time-dependence for formation, both strains formed biofilms (Figure 1 and Figure 2).

Most antimycotic drugs that disrupt yeast biofilms act in the initial stage of biofilm formation, preventing cell adhesion and subsequent biofilm consolidation [31]. In tune with this, Ctn[15–34] was able to inhibit biofilm formation and to eradicate the formed biofilms at concentrations 100 × the Ctn[15–34] MIC for the wild-type *Candida* strain. Drug diffusion and action are known to be restrained by biofilms [26,31] and, at this high peptide concentration (100 × MIC), the peptide caused only the inhibition of biofilm formation in the drug-resistant clinical isolate, without its eradication, despite the increased number of dead cells observed in the biofilm milieu (Figure 2 and Figure 3). As visible by AFM, these two *Candida* strains showed different phenotypes (Figure 2). Treatment with Ctn[15–34]-induced lethal morphological changes, as vesicle shedding, surface protruding, and bleb formation were evident in cells from the *Candida* biofilms exposed to 100 × MIC. Similar effects have been reported for *Candida* biofilms exposed to *Ps*d1, an antimicrobial peptide (defensin) isolated from *Pisum sativum* seeds [32].

The mechanisms by which Ctn[15–34] kills pathogenic fungi was, up to now, not fully understood, although membrane perturbation with disruption and leakage of intracellular content (e.g., lactate dehydrogenease) and apoptosis were shown to be involved in the activity against *Candida* [19,20]. Furthermore, detailed studies in bacteria indicated that the cytoplasmic membrane was the first molecular target for the microbicidal effect; Ctn[15–34] interacts with the plasma membrane of target cells by electrostatic forces, accumulates on the cell surface, and ultimately causes membrane permeabilization, resulting in rapid cell lysis. Secondary mechanisms, such as DNA binding, have also been invoked to reinforce the action of the Ctn[15–34] on bacterial cells [17].

In the absence of intrinsic Ctn[15–34] fluorescence, given the lack of tryptophan or tyrosine residues, a biophysical assay to verify the interaction of Ctn[15–34] and lipid membranes was done with the membrane probe di-8-ANEPPS. This probe is sensitive to changes in the membrane dipole potential, with its fluorescence spectra shifting in response to changes in the surrounding electric field, enabling the quantification of eventual binding of the peptide to the lipid membranes [33]. While no Ctn[15–34] interaction with lipid vesicles was detectable, the peptide-membrane interaction was indeed quantifiably evident when *C. albicans* cells were tested.

Dynamic light scattering was used to measure vesicle size before and after exposure to Ctn[15–34] (Figure 4). This technique relies on the principle of the Brownian motion, so that scattered light follows a pattern that depends on vesicle size [34]. Ctn[15–34], at 50 μM, induced a significant aggregation of fungal membrane-like lipid vesicles (Figure 4). To evaluate the influence of the fungal cell wall on the interaction of Ctn[15–34] and fungal cells, protoplasts of *C. albicans* were prepared and exposed to different peptide concentrations. Changes of zeta potential of *C. albicans* cells and protoplasts, as well as lipid vesicles exposed to Ctn[15–34], indicated that it interacts selectively with protoplasts and vesicles that mimic the fungal membrane (Figure 5). Therefore, it can be concluded that the *Candida* cell wall is not directly involved in the action of Ctn[15–34], but, importantly, the plasma membrane is the main target, playing a critical role on antifungal activity.

It is known that AMPs can cause changes in the topology of lipid membranes by interacting through distinct orientations (e.g., planar alignment parallel or perpendicular to the membrane surface, transmembrane orientation) [35]. For Ctn[15–34], the less negative zeta potentials observed with fungal membrane-like lipid vesicles and protoplasts suggest that the peptide was embedded in the lipid bilayer, contributing consequently to a less anionic surface charge and subsequent destabilization of cytoplasmic membrane with further membrane disruption and cell lysis. Moreover, the reduction in the surface charge of membrane-like lipid vesicles and protoplasts showed that Ctn[15–34] is capable of inducing membrane depolarization by a membrane composition-dependent process.

Overall, Ctn[15–34] is a multi-effector antimicrobial peptide with antifungal properties, with the fungal plasma membrane as first and main target, as demonstrated here by different biophysical and biochemical techniques. A scheme of this general behavior is presented in Figure 6. Ctn[15–34] can insert into the lipid bilayer and cause changes in membrane dipole potential. Moreover, Ctn[15–34] induces morphological changes in *Candida* cells within biofilms, causing their destabilization, although higher peptide concentrations are required, presumably to diffuse into the biofilm. The number of dead cells in the biofilm microenvironments increases along the time of exposure to Ctn[15–34]. Phenotypic differences between standard strain and fluconazole-resistant clinical isolate are indicative of the strain sensitivity and the biofilm stability of *Candida* exposed to the peptide action.

## 4. Material and Methods

### 4.1. Peptides

Ctn (KRFKKFFKKVKKSVKKRLKKIFKKPMVIGVTIPF), Ctn[1–14] (KRFKKFFKKVKKSV), and Ctn[15–34] (KKRLKKIFKKPMVIGVTIPF) were synthesized by solid phase methods, purified by HPLC to >95% homogeneity, and characterized by ESI-mass spectrometry, as previously described [13]. Peptide stock solutions (1 mM) were prepared in sterile Milli-Q water and stored in aliquots at −20 °C, until use.

### 4.2. Microorganisms

Two *C. albicans* strains were used: a wild-type standard strain (ATCC 90028) and a fluconazole (FCZ)-resistant clinical isolate (LABMIC 0125), provided by the Santa Casa de Misericordia Hospital in Sobral (Sobral, Brazil). The identification of the clinical isolate was performed using a CHROMagar-Candida (CHROMagar, Paris, France) and VITEK 2 automated identification system (BioMérieux, Lyon, France) with an YST card, and by PCR-AGE using the transcribed internal spacer (ITS) region.

### 4.3. Susceptibility of Planktonic C. albicans

*C. albicans* was cultured on Sabouraud-dextrose agar (SDA) and incubated at 35 °C for 24 h. Yeast colonies were transferred to tubes containing sterile phosphate buffer saline (PBS; 0.01 M Na_2_HPO_4_, 0.0018 M KH_2_PO_4_, 0.137 M NaCl, 0.0027 M KCl, pH 7.4) to obtain suspensions with a turbidity equivalent to Standard McFarland 0.5 (10^6^ CFU mL^−1^). These suspensions were then diluted 1:2000 in RPMI 1640 (with glutamine) medium/morpholinepropanesulfonic acid (MOPS), pH 7.4 [36]. All reagents were from Sigma (St. Louis, MO, USA).

The minimum inhibitory concentration (MIC) and minimum fungicidal concentration (MFC) of the peptides were determined by the broth microdilution method, using 96-well plates, according to the procedures of the Clinical and Laboratory Standards Institute (CLSI) M27-A3 standard [36]. *Candida* cells were exposed to peptide concentration ranging from 0.03 to 71.9 μg mL^−1^ (0.02 to 40 μM), in culture medium, in a microplate, incubated at 35 °C and fungal growth and/or inhibition was observed after 24 h. The MIC was defined as the lowest concentration where no visual growth (no turbidity) was observed, corresponding to 90% inhibition of fungal growth. The MFC was determined after the transference of 100 μL *Candida* suspension in which no turbidity was observed to capped test tubes containing SDA and incubated at 35 °C, for 48 h. The MFC was determined in accordance with fungal growth in the culture medium. Each experiment was performed in duplicate.

### 4.4. C. albicans Culture Preparation for Antibiofilm Assays

Stock cells were inoculated on 4% glucose SAD plates and incubated for 24 h at 35 °C. One colony was grown overnight in SDB and incubated at 35 °C with shaking at 180 rpm. Then, 100 µL of the overnight culture were diluted in 5 mL of SDB and incubated for 3 h with shaking at 35 °C. The cell concentration was measured by optical density at 660 nm.

### 4.5. Biofilm Development and XTT/Menadione Assay

For biofilm formation, RPMI 1640/MOPS supplemented with 2% glucose was used as culture medium. In this assay, wild-type *C. albicans* (ATCC 90028) and the clinical isolate LABMIC 0125 inocula were prepared as described above, but at cell suspensions of 10^4^, 10^5^, 10^6^, and 10^7^ CFU mL^−1^. Then, 100 µL of these cell suspensions were individually transferred to wells in a 96-well round bottom polypropylene plate. The plate was incubated for 12, 24, 48, 72, and 96 h, at 37 °C, with shaking at 180 rpm. At the end of the incubation period, the biofilm was washed three times with PBS to remove planktonic cells. Then, 100 µL of 200 μg of XTT/mL with 25 μM menadione was added to the wells of each plate and incubated for 2 h, at 37 °C, protected from light. The supernatants were transferred to a flat bottom plate and the optical density was measured at 490 nm [37].

### 4.6. Biofilm Inhibition and Eradication Evaluation by Confocal Microscopy

For biofilm inhibition and eradication, 8-wells µ-Slide assays plates (ibidi, Gräfelfing, Germany) were used. Cell concentrations and incubation times were determined by the biofilm formation assay described in 4.5, but with 10^4^ CFU/mL for *C. albicans* ATCC 90028 and 10^7^ CFU/mL for the clinical isolate, and an incubation time of 72 h. To perform the inhibition assays, 300 µL of each cell diluted in RPMI with 2% glucose was treated with Ctn[15–34] at the MIC, 10 × MIC, and 100 × MIC. For the eradication test, once the biofilm was formed, peptide was also added at the MIC, 10 × MIC, and 100 × MIC. After treatment, microplates were incubated for 24 h, at 37 °C. All plates were washed 10 × with sterile 10 mM HEPES, 150 mM NaCl, pH 7.4. To quantify peptide action on cells, the Live/Dead FungaLight Yeast Viability Kit (Life Technologies, Carlsbad, CA, USA) was used. Optical microscopy measurements with an inverted Zeiss LSM 710 confocal point-scanning microscope (Jena, Germany) were carried out in order to examine the architecture and the viability of *Candida* cells before and after exposure to antifungal agents. Argon (488 nm; 45 mW) and diode-pumped solid-state (561 nm; 15 mW) lasers were used with a 40 × dry-objective. After incubation, biofilms were stained with 2 µL SYTO 9 and propidium iodide (PI) dye probes (1:1) and incubated for 15 min. Images were acquired and analyzed with ImageJ 1.47v (rsbweb.nih.gov/ij/).

### 4.7. Atomic Force Microscopy Imaging

*C. albicans* biofilms were prepared as described for confocal microscopy assays (Section 4.6). After biofilm treatment with and without peptide, the cellular mass was carefully washed with Milli-Q water and air-dried at 25 °C. Images of untreated and treated yeast cells were conducted using a JPK NanoWizard IV atomic force microscope (Berlin, Germany) mounted on a Zeiss Axiovert 200 inverted microscope (Jena, Germany). Measurements were carried out in intermittent contact mode (air) using ACL silicon cantilevers (AppNano, Huntingdon, UK). Height and error images were recorded with similar AFM parameters (setpoint, scan rate, and gain). Scan rate was set between 0.3 and 0.6 Hz and setpoint close to 0.3 V. Images were analyzed with the JPK image processing software v. 6.1.149.

### 4.8. Fungal Protoplast Preparation

Protoplasts were prepared by removing the cell wall of *C. albicans*, which consist in a microfibrillar network composed by glucans and chitin embedded in an amorphous material composed mainly of mannoproteins that together provide rigidity to the cell wall [38,39]. Protoplasts were isolated with a yeast lytic enzyme (Zymolyase-20T, Grisp Research Solutions, Porto, Portugal), as described elsewhere [40]. Briefly, *C. albicans* (ATCC 90028) cultured in SDA was inoculated into yeast peptone glucose (YPG) broth and incubated overnight at 37 °C. After this period, cells were harvested by centrifugation at 3000× *g*, for 10 min, at room temperature. The pellet was washed with sterile Milli-Q water, resuspended, and incubated in pre-incubation medium (50 mM EDTA, pH 9.0, with 35 mM β-mercaptoethanol), for 30 min, at 37 °C, with shaking at 180 rpm. Cells was centrifuged and washed once with washing solution I (1.2 M sorbitol, 50 mM EDTA, pH 7.5). A second wash was carried out with washing solution II (1.2 M sorbitol, Tris 50 mM, pH 7.5). Cell concentration was determined by optical density at 660 nm. A suspension of 5 × 10^6^ CFU mL^−1^ was washed with enzyme buffer (1.2 M sorbitol, 50 mM Tris, 0.1 mM calcium acetate, 0.5 mM magnesium acetate). After washing, cells were resuspended in enzyme solution (enzyme buffer and 3 mg mL^−1^ Zymolyase-20T) and incubated for 2 h at 37 °C, with shaking at 180 rpm. Protoplasts were imaged with an optical microscope, by Gram staining, and maintained in HEPES buffer, at 4 °C, for up to 24 h after isolation.

### 4.9. Liposome Preparation

Large unilamellar vesicles (LUVs), with a diameter of approximately 100 nm, were obtained by extrusion of multilamellar vesicles (MLVs) as described elsewhere [41]. 1-Palmitoyl-2-oleoyl-*sn*-glycero-3-phosphocholine (POPC), 1-palmitoyl-2-oleoyl-*sn*-glycero-3-phosphoethanolamine (POPE), 1-palmitoyl-2-oleoyl-*sn*-glycero-3-phosphoinositol (POPI), and 1-palmitoyl-2-oleoyl-*sn*-glycero-3-phospho-L-serine (POPS) were from Avanti Polar Lipids (Alabaster, AL, USA), while cholesterol (Chol) and ergosterol (Erg) were from Sigma. The LUVs studied were mostly zwitterionic (pure POPC, POPC:Chol 70:30, POPC:Erg 70:30, and *C. albicans* membrane-like mixture, i.e., POPC:POPE:POPS:POPI:Erg 59:21:3:4:13 [42]), mimicking membrane systems relevant for the study [32,41]. Liposomes were extruded in the day of the measurement. HEPES buffer was used in all the measurements, except when otherwise indicated.

### 4.10. C. albicans Strains and Culture Conditions

*C. albicans* strains were maintained and grown as previously described [32]. From stock cultures kept at −80 °C until use, 10 µL of each strain were plated in SDA and incubated overnight, at 37 °C. An isolated colony was inoculated into 5 mL of SDB and allowed to grow overnight, at 37 °C, with shaking at 180 rpm. On the day of the measurement, 100 µL of the cell suspension were diluted in 5 mL of SDB and left at 37 °C until reaching log-phase, with a final cell counting of 10 × CFU mL^−1^. Then, cells were washed 3 × with SDB and centrifuged at 4000× *g*, at 10 °C, for 25 min. Finally, cells were suspended in HEPES or SDB, and reserved until use.

### 4.11. Fluorescence Spectroscopy Measurements Using di-8-ANEPPS

*C. albicans* at 10^5^ CFU mL^−1^ were incubated with 100 μM di-8-ANEPPS (4-(2-[6-(dioctylamino)-2-naphthalenyl]ethenyl)-1-(3-sulfopropyl)pyridinium inner salt) in HEPES buffer with 0.1% (*w*/*v*) Pluronic F-127 (Sigma, St. Louis, MO, USA), for 1 h, at 25 °C, with gentle agitation and protection from light. From this suspension, 10^4^ CFU mL^−1^ of cells and 10 μM of di-8-ANEPPS in a final volume of 500 µL were incubated for 1 h under stirring and protected from light. A suspension with the same concentration of unlabeled cells was used as control. Prior to the measurement, the suspensions were incubated with peptide (0–30 µM) for 1 h.

For lipid vesicle assays, LUVs were diluted in HEPES buffer to a final lipid concentration of 500 µM and labeled with 100 µM di-8-ANEPPS, for 1 h, under stirring in the dark. The same concentration of unlabeled vesicles was used as control. Labeled LUVs were incubated with peptide (0–30 µM) for 1 h.

Excitation spectra and the ratio of intensities were obtained at the excitation wavelengths of 455 and 525 nm (*R* = I_455_/I_525_), with emission at 670 nm. Excitation and emission slits were set to 5 and 10 nm, respectively. The variation of the ratio with the peptide concentrations was analyzed by a single binding site model using the expression
(1)$$\frac{R}{\;R_{0}}=1+\frac{\frac{R_{min}}{R_{0}}\cdot \left[Peptide\right]}{K_{d}+\left[Peptide\right]}$$
where *R* is divided by *R*_0_, the normalized ratio of intensities obtained without treatment with peptide, *R_min_* is the minimum asymptotic value of *R* and *K_d_* is the apparent dissociation constant [43]. Fitting of the equation to the experimental data was performed by non-linear regression with GraphPad Prism 6.

### 4.12. Dynamic Light Scattering and Zeta Potential Measurements

Dynamic light scattering (DLS) and zeta potential measurements were carried out on a Malvern Zetasizer Nano ZS device (Malvern, UK) and data was processed using the Malvern’s DTS software [44]. DLS was performed to assess possible changes induced by Ctn[15–34] in the size of LUVs. LUVs were diluted in 200 µM HEPES buffer, filtered through a 0.45 µM filter, and treated with 5, 10, or 50 µM peptide. An untreated control was also used. All measurements were conducted at 25 °C, using 1 cm optical path disposable cells, with 15 measurements (100 runs each) for each sample, after 15 min of equilibration time. Viscosity and refractive index were set at 0.8872 cP and 1.330, respectively. The results presented correspond to the average of three independent samples. Zeta potential measurements were performed with LUVs, and with *C. albicans* cells and protoplasts. LUVs were diluted in HEPES buffer at a lipid concentration of 200 µM and filtered through a 0.45 µM pore size filter. Vesicles were incubated for 1 h with 5, 10, or 50 µM of peptide prior to measurements. *C. albicans* and protoplasts were tested at 5 × 10^5^ CFU mL^−1^, in HEPES buffer. Applied current was set to 40 mV, with 60 s waiting between measurements.

## Figures and Tables

**Figure 1 ijms-21-08339-f001:**
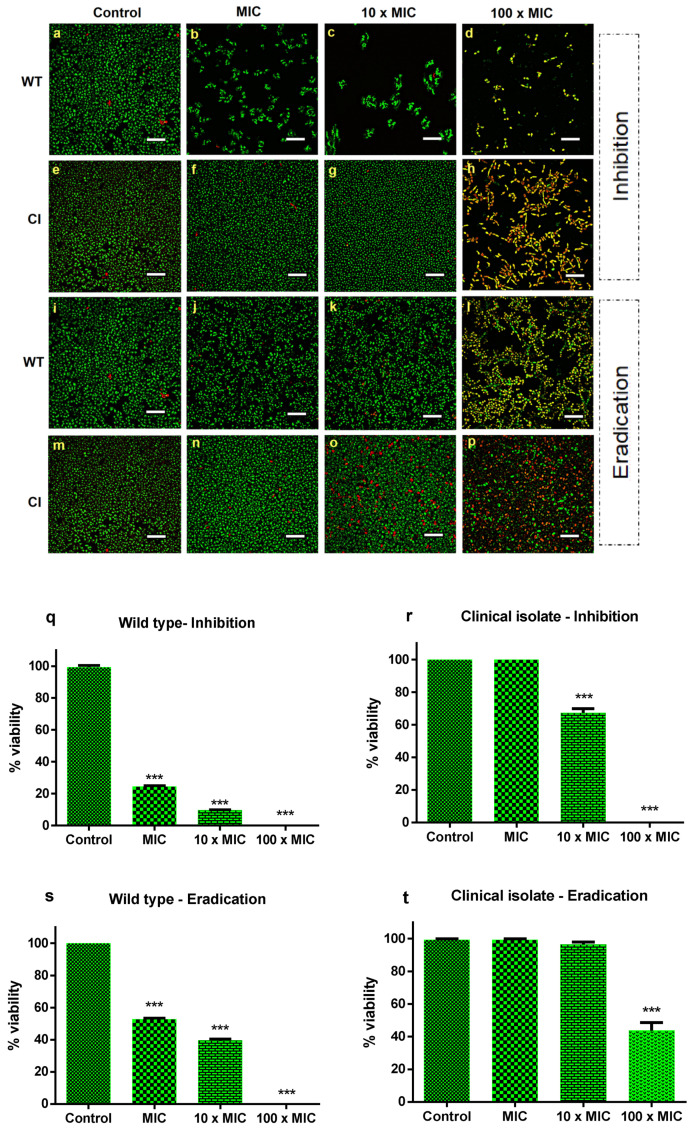
Effect of Ctn[15–34] on the inhibition and eradication of *C. albicans* biofilms. Confocal microscopy images of the inhibition of *C. albicans* biofilms for the sensitive wild-type strain (**a**–**d**) and the drug-resistant clinical isolate strain (**e**–**h**), and biofilm eradication of wild-type (**i**–**l**) and clinical isolate (**m**–**p**), in the absence (control) and presence of Ctn[15–34] at different concentrations. Images obtained with live/dead staining (SYTO 9, green; PI, red) using 40-fold magnification. Scale bars: 30 µM. Quantification of viable cells (**q**–**t**) obtained using the ImageJ software. Bars represent mean ± SD. *** *p* < 0.001 relative to control (without peptide).

**Figure 2 ijms-21-08339-f002:**
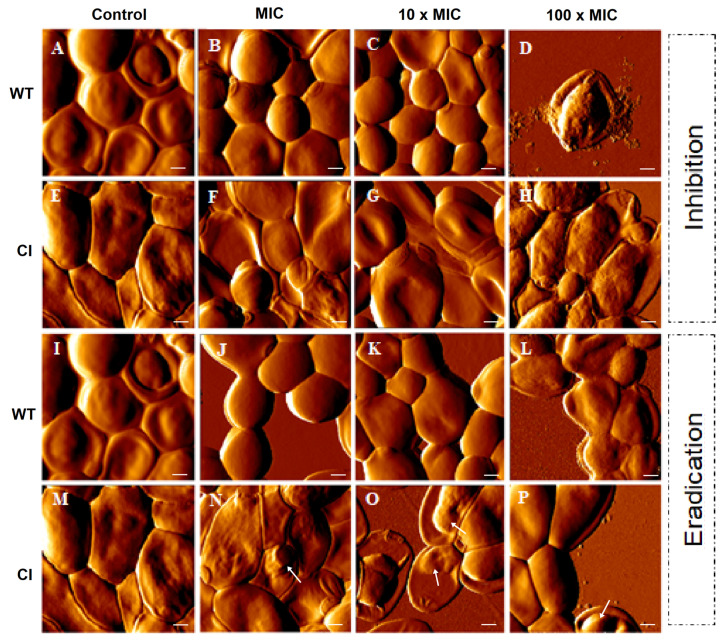
AFM images of *C. albicans* biofilms. Formation (control) (**A**,**E**,**I**,**M**), inhibition, and eradication assays were performed for both strains (wild-type and clinical isolate). Images show the effect on *C. albicans* of the antimicotic peptide at different concentrations (MIC, 10 × MIC, and 100 × MIC) (**B–D**, **F–H**, **J–L**, **N–P**). Images of formation of biofilm represent the control without peptide. Total scanning area for each image: 10 × 10 µm^2^. Vesicles on the cell surface (**N**–**P**) were identified using white arrows. Scale bars: 1 µM.

**Figure 3 ijms-21-08339-f003:**
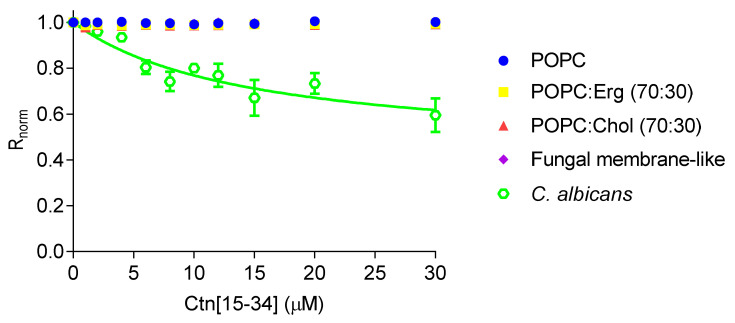
Interaction of Ctn[15–34] with lipid vesicles and *C. albicans* protoplasts, assessed with the membrane dipole potential-sensitive probe di-8-ANEPPS. The excitation ratio, *R* (I_455_/I_525_), was normalized by dividing by the *R* value obtained in the absence of Ctn[15–34]. Equation (1) was used to fit the experimental data. Data are presented as mean ± SD of three independent measurements.

**Figure 4 ijms-21-08339-f004:**
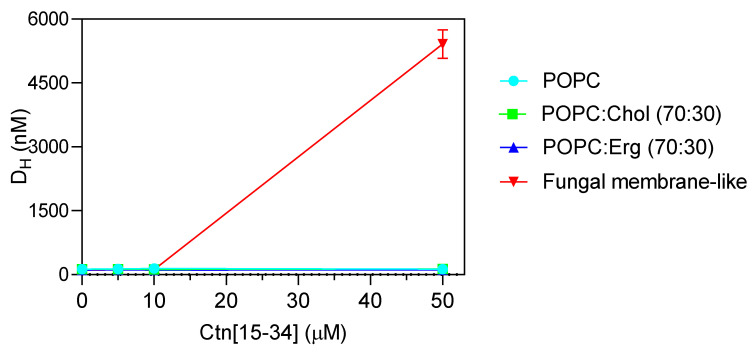
Hydrodynamic diameter (D_H_) of the different lipid vesicles (200 µM total lipid concentration) exposed to increasing concentrations of Ctn[15–34]. Extensive aggregation is clear at the concentration of 50 µM for the fungal membrane-like lipid composition. Data are presented as mean ± SD of three independent replicates.

**Figure 5 ijms-21-08339-f005:**
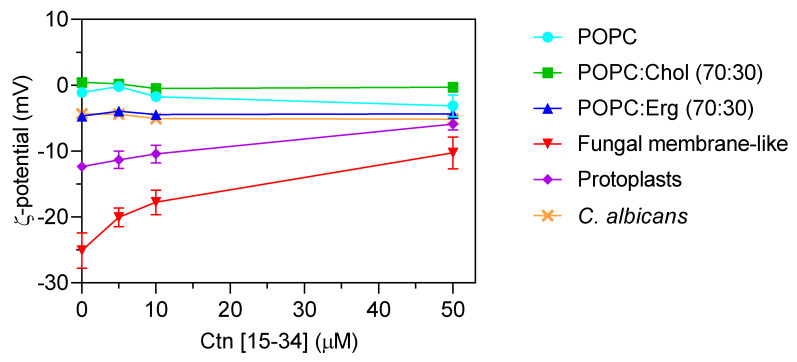
Zeta potential (ζ) of lipid vesicles, *C. albicans* protoplasts, and *C. albicans* cells. Lipid vesicles were tested at a total lipid concentration of 200 µM. *C. abicans* cells and protoplasts were tested at the concentration of 10^5^ cells mL^−1^. The results showed the importance of some membrane components of the cell membrane for Ctn[15–34] interaction. *C. albicans* and POPC:Erg vesicles show similar zeta potential behavior upon increasing Ctn[15–34] concentration. The initial values are −12 mV for protoplasts and –25 mV for fungal membrane-like vesicles. Data are presented as mean ± SD of three independent replicates.

**Figure 6 ijms-21-08339-f006:**
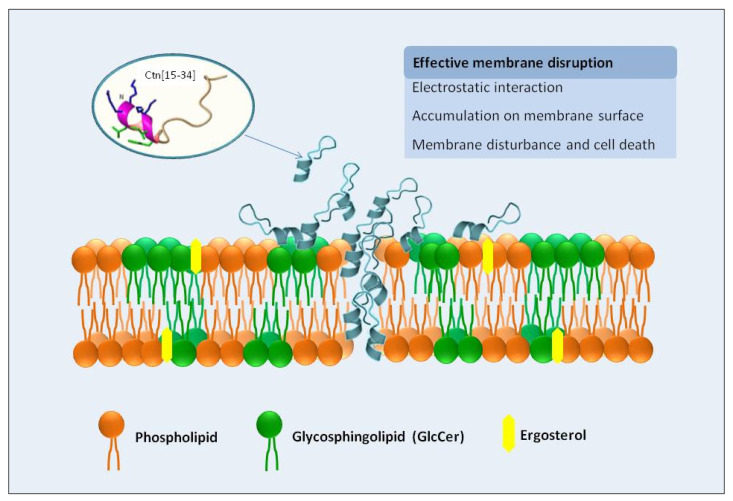
Depicted mechanism of action of Ctn[15–34] on the interaction and disruption of plasma membrane of the pathogenic yeast *Candida albicans*. The plasma membrane of yeast cells is the primary target of Ctn[15–34] actions, as demonstrated experimentally by biophysical methods. Ctn[15–34] is recruited to the plasma membrane, through a process facilitated by the electrostatic attraction between the cationic residues of the peptide and the anionic lipids present in the plasma membrane of planktonic yeast or yeast cells in biofilms. After accumulation on the cell surface, peptide can efficiently insert into the lipid bilayer, causing membrane destabilization, as observed by changes in membrane dipole potential, with consequent death of the *C. albicans* cells.

**Table 1 ijms-21-08339-t001:** Minimum inhibitory concentration (MIC) of Ctn[15–34] against *Candida albicans*.

Strain	Source	MIC, µM (µg mL^−1^)	
		Ctn[15–34]	Fluconazole
ATCC 90028	Culture collection	10 (23.7)	3.12 (1)
LABMIC 0125 *	Deep tissue injury	10 (23.7)	25 (8)

* Resistant strain according to the breakpoints of the European Committee on Antimicrobial Susceptibility Testing (EUCAST) and Clinical & Laboratory Standards Institute (CLSI), as described in materials and methods, which establish MIC > 4 µg mL^−1^ for strains of fluconazole-resistant *Candida albicans*.

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
