# Peer review of "Antibiofilm Activity on Candida albicans and Mechanism of Action on Biomembrane Models of the Antimicrobial Peptide Ctn[15–34]"

_ijms, 2020, doi:10.3390/ijms21218339_

Round 1

Reviewer 1 Report

The additional descriptions and supplemental images have greatly improved this manuscript. 

Author Response

Thank you very much for all your suggestions.

Reviewer 2 Report

Overall, the research design, presentation, and interpretation of the experimental findings, was solid. Although the paper was generally well-written, there were several typographical, syntactical, and contextual revisions that should be made prior to accepting the final draft.

Specific comments are below.

The C. albicans reference strain (ATCC 90028) was described as “the sensitive strain” tor standard strain throughout the manuscript body, and “wild-type” was used in the figure labels for figures 1 and 2. “Sensitive” is usually used to describe C. albicans mutant strains.  For consistency, the authors should use describe the reference strain as “wild-type” throughout the manuscript.  

Lines 116-117 and lines 251-252. The authors state that the wild-type and resistant strains formed biofilms at the same rate but different cell concentrations; however, the basis for this was not addressed. It should be explicitly stated that the cell and matrix density in the mature 72 hr biofilm is the same between both strains, so that differences in Ctn15-34 biofilm inhibition is not due to AMP accessibility within the biofilm.

Lines 170-172, 261, and Figure 2. The authors state that small vesicles are visible on the surface of drug-resistant cells. These vesicles are not apparent, nor is the evidence convincing that the aforementioned image contains vesicles.  The figure should include an arrow pointing to these vesicles, and if appropriate, the authors should state that they cannot rule out the possibility that the vesicles may be another cellular structure. Otherwise, they would need to prove (i.e. EM immuno-microscopy) that these structures are actual vesicles. Lastly, the text references figure 2L which is the image for the wild-type strain instead of figure 2p.

Lines 285-288 “Protoplasts were prepared by removing the cell... cell wall [36,37].” These sentences can be deleted from the discussion, as they do not provide insight, and this information is common knowledge in the mycology community.

Lines 302-310. A schematic model displaying Ctn15-34’s mechanism-of-action on membranes would strengthen the manuscript and should not be labor-intensive to produce.

Sections 4.4 - 4.7. The degree symbol is underlined. This was not observed in sections 4.1-4.3 where the symbol was correctly written without being underlined.

Line 361. The authors state the biofilm assay conditions were described in section 2.4. This should be changed to section 2.1, as section 2.4 is not in the manuscript.

Author Response

Answers to Specific comments are below.

  1. The C. albicans reference strain (ATCC 90028) was described as “the sensitive strain” tor standard strain throughout the manuscript body, and “wild-type” was used in the figure labels for figures 1 and 2. “Sensitive” is usually used to describe C. albicans mutant strains. For consistency, the authors should use describe the reference strain as “wild-type” throughout the manuscript. As suggested by the Reviewer, reference strain was renamed to “wild type” throughout the manuscript.
  2. Lines 116-117 and lines 251-252. The authors state that the wild-type and resistant strains formed biofilms at the same rate but different cell concentrations; however, the basis for this was not addressed. It should be explicitly stated that the cell and matrix density in the mature 72 hr biofilm is the same between both strains, so that differences in Ctn15-34 biofilm inhibition is not due to AMP accessibility within the biofilm. As suggested by the Reviewer this two lines (257-258) were added to the text: As observed, despite the cell number- and time-dependence for formation, both strains formed biofilms (Figure 1 and 2).
  3. Lines 170-172, 261, and Figure 2. The authors state that small vesicles are visible on the surface of drug-resistant cells. These vesicles are not apparent, nor is the evidence convincing that the aforementioned image contains vesicles. The figure should include an arrow pointing to these vesicles, and if appropriate, the authors should state that they cannot rule out the possibility that the vesicles may be another cellular structure. Otherwise, they would need to prove (i.e. EM immuno-microscopy) that these structures are actual vesicles. Lastly, the text references figure 2L which is the image for the wild-type strain instead of figure 2p. As suggested by the Reviewer, an arrow was included in Fig. 2 pointing to the vesicles and lines 175-176 was modified as follow: “show apparently small vesicles on the surface of the drug-resistant cells when exposed to the highest concentration of Ctn[15-34]”.
  4. Lines 285-288 “Protoplasts were prepared by removing the cell... cell wall [36,37].” These sentences can be deleted from the discussion, as they do not provide insight, and this information is common knowledge in the mycology community. As suggested by the Reviewer, the sentence was deleted.
  5. Lines 302-310. A schematic model displaying Ctn15-34’s mechanism-of-action on membranes would strengthen the manuscript and should not be labor-intensive to produce. As suggested by the Reviewer, a schematic representation of the mode of action was included in the manuscript as Figure 6.
  6. Sections 4.4 - 4.7. The degree symbol is underlined. This was not observed in sections 4.1-4.3 where the symbol was correctly written without being underlined. As suggested by the Reviewer, this misspelling was corrected along the manuscript.
  7. Line 361. The authors state the biofilm assay conditions were described in section 2.4. This should be changed to section 2.1, as section 2.4 is not in the manuscript. As pointed by the Reviewer this misspelling was corrected along the manuscript.

Reviewer 3 Report

Overall excellent manuscript. The experimental plan was clearly defined and results well explained, especially for a biologist with little experience in materials physics. 

Author Response

Thanks you very much for all your suggestions.

This manuscript is a resubmission of an earlier submission. The following is a list of the peer review reports and author responses from that submission.

Round 1

Reviewer 1 Report

The work concerns the important problem of searching for effective anti-biofilm agents that prevent the formation of the fungal biofilm or allow its destruction.  It takes into account a peptide that is the C-terminal fragment of crotalicidin, previously selected as an anti-fungal compound. The authors  attempt to demonstrate the effective action of this peptide and discover the mechanism of its action on fungal biofilm. Despite the use of advanced methods, the problem of the mechanism remains unsolved, see the comments  below:

1/ the standardization of the starting cell number presented in  Fig.1 should be presented as supplementary material;

2/ the statement “the activity of Ctn[15-34] on the inhibition of biofilm formation” should be change to “the inhibitory activity of Ctn on biofilm formation”;

3/ the description in Figure 2 (a-p) does not correspond to its description within the text - see the sentences: “The peptide effect on biofilms were visualized with confocal microscopy (Figure 2A)”, “ This behavior was observed both in inhibition and eradication assays (Figure 2B)”;

4/ line 121 “whereas in the resistant strain there was no change in cell mass” - this statement is not true as the authors did not make any cell mass determination;

5/ is the image from a “confocal microscope using 40-fold objective magnification” actually presented in Fig. 2?

6/ Fig 2 and 3 should be combined, as Fig 3 is only an interpretation of the data presented in  Fig.2;

7/ it is not clear why the authors used different temperatures (37 and 35°C) for fungal cell propagation and why did they not observe any hyphae for the low cell density;

8/ I disagree that the authors presented any mechanism of Ctn[15-34] action; instead, they acquired a set of uncorrelated data:

a/ using staining with di-8-ANEPPS (no explanation for the abbreviation in the text) and titration with the peptide, the authors determined a dissociation constant (Kd) of 14.5 ± 5.6  (no unit provided) – but they do not comment on it in the discussion. If the authors only wanted to show the interaction of the peptide with C. albicans and its lack in contact with the lipid membrane, they should not have determined the Kd value, especially with the 1: 1 interaction model. Moreover, the conclusion from this assay indicates an interaction of the peptide with the components of the yeast cell wall;

b/ the data in the Fig 6A (not 5A as is described in the text) presenting the changes in hydrodynamic diameter (DH) is not well composed as no changes of DH were observed in the MIC range, important for the effective action of the peptide on biofilm (10-50 microM). All further conclusions are based on the single experimental point (!) suggesting peptide aggregation on the fungal membrane-like vesicles.   By the way - the presentation of fig. 6B is not necessary;

c/ the most controversial are the results concerning Zeta potential of lipid vesicles, protoplast, and C. albicans cells. Positive changes were observed only for protoplast and fungal membrane-like vesicles, without any changes for C. albicans, and the authors in the discussion concluded that  “Candida cell wall is not involved in the action of the peptide but, oppositely and importantly, the cytoplasmic membrane plays a critical role”. The results did not correspond to the data presenting peptide binding on the fungal cell surface. The possible translocation of the peptide to the cell membrane should be proven;

9/ “Indeed, drug diffusion and action are restrained by microbial biofilms – a virulent phenotype” – biofilm is not the cell phenotype, see the literature (Fanning S, Mitchell AP (2012) Fungal Biofilms. PLoS Pathog 8(4): e1002585) – “Biofilms are complex surface-associated cell populations embedded in an ECM that possess distinct phenotypes compared to their planktonic cell counterparts”

10/ the text requires language corrections

In conclusion, despite the use of advanced research methods, the authors only proved that the peptide is effective against fungal biofilm formed by different C. albicans strains at a much higher concentration than for the single yeast cells. Further conclusions require better experimental support, e.g. using labeled peptide.

Reviewer 2 Report

An antimicrobial peptide, Ctn[15-34], derived from rattlesnake venom has previously been demonstrated to be antibacterial, antiparasitic and antiviral. The efficacy of this antimicrobial peptide also has been previously demonstrated on Candida albicans. However these previous studies were conducted on planktonic cells. The unique focus of the study described in this manuscript is to demonstrate the efficacy of Ctn[15-34] on C. albicans biofilms sensitive and resistant to antifungals chemotherapeutics. Additionally, this study is mechanistic by providing evidence for breakdown in the integrity of the cell membrane and postulating that the peptide is intercalating in the lipid membrane. 

  1. The data in figure 2 and the quantitation of the images (by the use of software) in figure 3 is the evidence that the authors use to support the efficacy on inhibiting biofilm formation and eradicating established biofilms. These figures appear to be singular confocal planes and not z-stacked. C. albicans is dimorphic and under the conditions for biofilm inhibition (37oC in RPMI) I would expect to see hyphae with in the biofilm.
    • Which leaves me to ask are you looking the viability of the complete biofilm in figure 1 (especially at MIC and 10XMIC concentrations?
    • Also in figure 3 it appears that viability is based on biofilm density.  Figure 1O represents CI at 10X MIC and you can clearly see more red staining compared to wild type in Figure 2O indicating that the cell membranes are compromised. However the quantitation of these images show there is no change compared to untreated or MIC levels with narrow error bars. But comparing all the panels in figure 2 and 3, it appears that a reduction in the number of cells is associated with the calculation of viability. The data could also indicate that the peptide is affecting cell adhesion and not viability alone.
    • The residual biofilms in Figure 2B and C. are captured while the cells are moving. This is further indication that cells in the biofilm are not adherent upon peptide treatment and still alive given that they are not red and perhaps you are losing biofilm mass. It could also indicate that there was some vibration when capturing the images.
    • Strengthening the Data: 
      • As the data stands adhesion and viability can not be separated out.
      • The authors use XTT assays to asses the quality of biofilms but do not show results with peptide treatment. XTT assays with minimal washes may strengthen the quantitation of their confocal data.
      • Confocal data as the only experiment to support viability. A rendered z stack images of the biofilm would be helpful to get a better idea of the whole biofilm especially at lower MIC and MICX10 concentrations.
      • Further description of how viability is quantified and calculated in figure 2 would also be helpful. It is unclear if they using the red or yellow fluorescence for their calculations. 
  2. Biofilms are transferred to 35oC for treatment with peptide during the eradication assays. Why are the biofilms treated at this temperature? Does this the hyphae in the biofilm?
  3.  Figure 4 arrow indicators maybe helpful to direct the eye of the reader.

Minor edits

Line 55 the use of the word "educated" in this sentence is unclear.